# Exploring Teacher Education for Sustainable Development in the UAE

Sandra Baroudi

College of Interdisciplinary Studies, Zayed University, Dubai 19282, United Arab Emirates;
sandra.baroudi@zu.ac.ae; Tel.: +971-502836642

**Abstract:** New ways of modernizing professional development programs (hereafter PDs) focus on the acquisition of new teaching methods and techniques through hands-on opportunities provided to teachers, thereby enabling them to practice and reflect on the knowledge gained. Moreover, the new vision of reform and sustainability in education emphasizes the development of sustainable PDs that resist disruptive factors, increase teachers' commitment, and ensure more sustainable development across children's lifespans. This study explores the impact of a piloted sustainable PD on the development of 16 teachers and head teachers' attitudes about assessment practices, professional growth, and leadership skills. Qualitative data collected in the form of interviews, post-PD surveys, and reflections were analyzed using a thematic analysis approach. Findings indicate that the confidence in creating assessments and rubrics of participants in their different positions increased, and their ability to provide their students with engaging, sustainably developed assessments that improved higher-order thinking skills was also enhanced. The findings in this study demonstrated the need to develop within teachers the awareness that they can individually contribute towards a more sustainably developed classroom and learning environment. Although participants did make some changes to their classroom-based practices, these changes could not be well-maintained as they were limited by the high stakes involved in varying the structure of mandatory assessments required in the public education sector. It is hoped that the findings of this study can be used as a model for the development of sustainable PDs in education.

**Keywords:** sustainable professional development; classroom-based assessment practices; teachers' attitudes and beliefs; teachers' professional growth

## 1. Introduction

The UN's 2030 agenda for sustainable development calls for renewed thinking to achieve peace, prosperity, and equitable experiences for humanity [1] (p. 14). Education is one central domain that can contribute to meaningful, sustainable development that will ensure a changed and better experience in knowledge acquisition for generations to come [2]. This paper calls for the development of sustainable PDs for teachers as a way to not only empower their practice but easily introduce sustainable development in their teaching to ensure inclusive and equitable quality education and promote lifelong learning for students [3,4].

In reality, traditional PDs do not reflect teachers' desires; they are passive experiences leading to no change in everyday practices. By reflecting on teachers' desires and agentively involving them in the development process, the PD begins to become meaningful and more sustainable for all stakeholders [5]. Critically, merely altering what teachers know or simply adding to their knowledge is not enough to effect change and ensure sustainable development; teachers must practice the methods in the classrooms; and through this down-up trajectory, teachers can enhance the learning experience of children and influence change at the policy level. To this end, researchers like [6] recommended modernizing training methods by avoiding excessive theoretical knowledge and focusing more on acquisition of

new teaching methods and techniques. Teachers participating in those PDs were offered new roles and tasks in the school, which had positive influences on their personal (i.e., self-efficacy, leadership skills) and professional growth (i.e., became more aware of their role as teachers) [7]. The training described in this paper offered teachers autonomy and agency in their own self-development, and by doing so ensured more sustainable development in teaching, learning, and assessment methods.

Following the pandemic, the UAE initiated a large-scale PD project to train government schoolteachers and meet the expectations of the 2030 vision [8]. A series of PDs were identified based on school teachers' needs. Assessment and evaluation of student learning was one of these highlighted needs. Consequently, a professor working at the federal higher education institution in the UAE was asked to design and deliver a sustainable PD module to improve the assessment skills of teachers and head teachers at the National Charity Schools (hereafter NCS) over a period of 25 h spread over a period of 10 weeks. Accordingly, this PD was designed on pre-defined criteria to ensure sustainability, continuity, resistance to disruptions, and improvement in teachers' commitment and performance [7]. The length of the PD was decided based on synthesis of the evidence from a systematic review and meta-analysis [9], which stated that PDs of less than 30 h had a significant effect on student learning outcomes in reading (g = 0.367, $p < 0.001$) when compared with PDs above 30 h. Shorter PDs help teachers focus more on condensed elements and specific strategies and resources that they could implement in class—unlike longer PDs in which teachers take longer to implement what they learned, thereby possibly causing delays in reporting the evidence on students [9].

Thus, the purpose of this study is to explore the impact of this PD program on changing schoolteachers' mindsets and practices towards classroom assessments as a means to improve student learning outcomes and implement elements of effective sustainability in education.

## 2. Literature Review

Effective PDs are structured professional learning programs that aim to improve teaching practices and increase student learning outcomes [10–12]. To that end, it is recommended that teacher beliefs be aligned with the values of the profession for the training to be effective [13]. It is also recommended that the PDs be designed based on teachers' diagnosis of their needs and to consider the complete set of parameters, such as content, duration, the trainer, along with the incentives and resources of the program [14].

Traditional PD has recently been in decline internationally and gradually pivoted to a new perspective of viewing teachers as active learning participants through reflection [15]. Experiential learning is not a novelty in PD and during implementation, teachers' experiences in developing practices in action go through the cycle of being introduced to professional training followed by experimenting, reflecting, and adapting [16]. Since personal growth is a part of PD, reflective practice is often the means to attain such growth [9]. This direction towards reflective and active practices through PD has been theoretically underpinned by experiential learning, which entails different learning approaches, including problem and inquiry-based learning, all of which are rooted in Piaget, Dewey, Hahn, and Vygotsky [16]. Just as change is a gradual process, so is the development of new practices [17]. Consequently, at the center of each practice lies a concrete experience that the learner can think about, reflect on, and act on accordingly. Experiential learning design requires experimenting, reflecting on the experimentation and experience, and drawing abstraction from the reflection and its resulting action [16]. The sustainable approaches of PD inspired by experiential learning and classroom-based have shown positive results in developing practices and changing personal beliefs [18].

Any PD should support teachers' personal growth and development, as well as their teaching practices and beliefs [10]. Most of the studies on professional development focus on teacher morale and sense of recognition as a means to improve their performance [19]. The best PD programs are the ones that associate ongoing practice-based inquiry with

classroom-based coaching [20]. According to [21], students' achievements are likely to improve when schools promote ongoing teacher-centered, collaborative, and research-based learning activities. Studies show that to achieve high quality education, a PD must be continuous so that teachers can reflect on their practices and tune into how to improve their competences and ultimately those of their students [10,22]. Studies to date show evidence of the positive relationship between teacher engagement in PDs and student outcomes [21,23]. Particularly in low-and middle-income countries, PDs are the main means to refining teachers' effective teaching skills as they develop their attitudes about the subject matter and hone their competencies and skills [14]. It was also proven that the creation of a positive school culture, citizenship, and development of opportunities for peer learning are also outcomes of good PDs. In particular, teachers receiving extensive PDs, where they have the opportunity to practice with other teachers, are highly correlated with student learning [14].

PDs benefit teachers' personal growth and leadership skills; however, not many studies have been conducted regarding how teachers become leaders and how PDs should be designed [24]. To this end, it is necessary to understand the process by which teachers grow professionally and conditions that support and promote that growth when designing PD programs [25]. As such, successful PDs are those that not only help teachers acquire new ways of thinking about learning, learners, and subject matter but also promote teachers' confidence in taking on new leadership roles [13,25]. Moreover, a teacher-centered approach to leadership focuses on individual decisions that teachers make. Evaluations of what does or does not represent their preferences will be highly individual, and personal agendas, goals, and priorities are personalized. In short, the 'one-size-fits-all' concept does not work. Thus, teachers will rarely be fully engaged with new forms of practice unless they see them as potential improvements to existing practice. Such commitment involves attitudinal and/or intellectual change, albeit sometimes in very small measures.

There is a consensus among researchers that teachers' beliefs about their role greatly influence their teaching effectiveness, and thus student learning and achievement [13]. For instance, in a recent study conducted by [26], 118 Turkish EFL pre-service teachers stated that online education during the pandemic ensured the sustainability of education only when they were prepared to design and implement interactive online lesson plans and group activities for students to increase students' engagement and speaking skills. This study concluded that teachers' effectiveness is associated with the effectiveness of the online education and ensuring that no child is left behind the screens. To that end, the best PDs to refine teachers' beliefs and improve their instructional practices are the ones that provide incentives to teachers and help them become agents of change [25]. Teachers' attitudes change once they have made a change in their classroom practices and seen improvement in student learning [27]; the students' outcome motivates and encourages teachers to foster practical change [16]. Thus, strong PDs have a positive effect on teacher efficacy resulting in an empowering change factor [28,29].

Furthermore, teachers' values and conceptions of teaching seemed to influence their assessment policies. Teachers assimilated new assessment practices only if they changed their notions of assessment [30]. Similarly, for a new policy on assessment to succeed and innovation in assessment practices to occur, teachers' conceptions must be taken into consideration [30]. Teachers' conceptions of assessment directly impact how they teach and what students learn [30,31]. Notably, the teacher choices of assessment techniques are directly linked to their beliefs about student self-confidence, morale, creativity, and work [30]. Hence, it is suggested that new assessment policies to enhance student learning should aim to maximize the link in teachers' minds regarding their commitment to improving the teaching/learning process [30]. In other words, the bettering of student assessments would possibly lead to improved quality in student learning outcomes. Consequently, the development of assessment policies should first identify teachers' conceptions of assessments.

With the continuously demanding educational requirements to meet diverse students' needs while offering equity in twenty-first century learning, teachers, especially novice ones, struggle to differentiate their instruction and plan assessments accordingly. Thus, there is a pressing need to train teachers in how to reach the goal of differentiating their instruction based on student needs [32]. A study found that teachers' understanding and implementation of formative assessment in their classes proved to be vague or ambiguous as they could not distinguish between formative and summative assessments; hence, they were unable to provide effective useful feedback [33]. Teacher trainees participating in the study showed improvement in their feedback implementation, especially that twenty-first century skills needed a new range of competencies essential for citizenship, social duties, and personal fulfillment [33]. In fact, policymakers and educators realize perfectly well that the quality of feedback drives the usefulness and value of assessments, thereby driving effective student learning [34]. Additionally, developing higher-order thinking skills is a priority in the twenty-first century skill-building framework, and integrating these skills into school-based assessments is essential. However, teachers whose knowledge of higher-order thinking skills seems to be mediocre have not shown they are competent in providing effective and quality assessments [35]. Consequently, there is a strong need to provide constant support and monitoring of teachers' assessment practices, and practical and effective professional development focusing on guiding methods for integrating higher-order thinking skills in teachers' assessments [36]. To compensate for the lack of teachers' adequate competencies in the new requirements of the teaching and learning process (especially after the recent COVID-19 pandemic), it is equally important to organize up-to-date and effective professional development, as well as lifelong learning support for teachers [6]. To this end, designing sustainable PD programs based on teachers' needs would stimulate and inspire teachers to make substantial changes to their everyday practices and behaviors, and allow them to reflect on and observe the impact of these changes on their teaching effectiveness and students' outcomes [6], and consequently rekindle their passion for teaching [14].

The Ministry of Education in the UAE has placed great value on training teachers because they know that empowerment of teachers is central to educational reform [8]. The UAE encouraged the design of sustainable PDs, especially during the COVID pandemic. Sustainability is when a program is capable of adapting to change by using its core beliefs and values as a guideline to resist disrupting factors [37]. To create such a sustainable PD to which teachers are committed, PD designers must consider that the PD ensures (1) opportunities for teachers to become communities of practice, (2) an increase in teachers' leadership skills, (3) recognition of teachers' performance, and (4) expanding teachers' professional orientation [7]. To make use of this vision of sustainability, widely known by major educational institutions and the United Nations, effective and practical PDs must be created to empower teachers in meeting these challenges [38]. To that end, a new PD module focused on assessment practices was designed based on these criteria and delivered to teachers at the NCS in the UAE.

## 3. Materials and Methods

### 3.1. Design

This is a qualitative case study design aimed at collecting detailed information from participants using different data collection procedures over a period of time and providing rich descriptive analysis of participants' perceptions about the impact of the PD module on their attitudes, leadership skills, and professional development [39]. This design was adopted in particular because of the uniqueness of the module and participants, as the PD was developed specifically to meet the needs of teachers and head teachers at the NCS in the UAE. As such, the main purpose of this qualitative research approach is to allow for deeper comprehension of the phenomenon through exploring participants' perspectives in a real-life setting [40,41].

### 3.2. Participants

The population in this study consisted of teachers ($n = 9$) and head teachers ($n = 7$); nine participants were females and seven were males. The participants were purposefully selected to participate in this study. They taught different subjects (science ($n = 7$); Arabic language ($n = 4$), English language ($n = 2$), and Math ($n = 3$)). The ethnicity of the participants was Arab with various nationalities: seven Jordanians, four Egyptians, three Syrians, one Palestinian, and one Algerian. Among them, 44% were between 30 and 39, 31% 40–49, 19% 50–59, and 6% above 60 years old. Twelve percent of participants had more than 15 years of teaching experience and 38% had worked at the NCS for over 10 years. On education level, 81% of the teachers and head teachers held a Bachelor of Science and/or a teaching diploma, 13% held a Master's degree, and 6% held a PhD in education. The seven head teachers were not only teaching different subjects but were also responsible for leading the department including supporting teachers, creating the lesson plans and assessments, monitoring the teaching and learning process, and supporting teachers by recommending them to participate in PDs based on their needs.

### 3.3. Instruments and Ethics

A variety of data collection tools were used in this case study. For the PD, two faculty colleagues at the same federal university with more than 10 years of experience in teaching and training examined the design of the PD: its content, materials, activities, and tasks (see introduction above). The PD was designed based on two modalities; face-to-face (at one of the school branches) and online to increase teachers' attendance and participation and be ready for possible disruptions. Furthermore, it gave participants opportunities to practice what they learned during and after each session. Teachers and head teachers were engaged in peer and group activities during the sessions and were asked to implement what they learned in their assessment practices. They were also asked to submit a weekly reflection about the changes (if any) they implemented and reflect on the impact of these changes on their practice. This PD ensured the recognition of teachers' participation through the attested certificates they received at the end of the module. The learning outcomes of this module aimed at supporting teachers and head teachers to:

1. Reflect on the different purposes, modes, and types of assessment.
2. Recognize the importance of instructional assessment as assessment-for-learning and the role of feeding back/feeding forward in effective teaching and learning.
3. Demonstrate the ability to effectively use the assessment to engage students and create deep learning.
4. Design and develop formative and summative assessment tools with rubrics that align with target learning outcomes.
5. Analyze and interpret assessment data.

First, the researcher invited all teachers and head teachers who participated in the module for individual online interviews. Seven participants agreed to participate in the interview. At the beginning of the interview, participants consented that this study use the data collected through the interviews. The researcher assured participants anonymity of identity and confidentiality of data shared. The virtual interviews were recorded on Zoom and transcribed manually. Each interview, lasting 30 min, included 18 open-ended questions developed according to the themes identified in the literature i.e., [10,13,14] and related to the impact of professional development programs on teachers' attitudes and beliefs, as well as on their professional and personal growth. Interview questions were designed to gain insight about the extent that the module increased participants' (a) knowledge and skills in assessments, (b) confidence in assessing students' learning, (c) personal growth, such as leadership opportunities within school, and (d) practical application of knowledge gained in their instruction to improve student learning. See Appendix A for interview questions. The interview questions were piloted with two participants at the beginning of the study to ensure that they are clear and understood the same way by both of them.

Second, during the last session of the PD module, participants filled out a survey to collect their feedback about the sessions. The use of this second tool was important to increase the validity of the findings and reduce inaccurate conclusions [42]. The first part of the survey included eight questions to collect participants demographics; the second part included 19 questions based on a 5-Likert scale ranging from very satisfied to very dissatisfied; and the third part constituted three open-ended questions to understand the challenges, if any, that the participants faced regarding the location of the PD, the timing, the topics of the sessions, and their recommendations. See Appendix B for survey questions. The survey was piloted to a random sample of five teachers and headteachers to check its reliability and validity and ensure that the wording of the questions was clear and free of jargon. Hence, two questions (number 12 and 18) were reworded to guarantee that the meaning of the questions was accurate and uniformly understood by all participants.

Third, as this PD module was designed based on sustainability criteria such as giving the chance for participants to practice and exchange expertise [7], it was essential to engage participants in the practice of reflection. Hence, after each session, participants were asked to identify the main point discussed in the session, analyze what was discussed in regard to this main idea, identify some changes they could make to improve their assessment methods, implement these changes in the classroom, and reflect on the process and results. A total of 128 reflections were collected from participants to ensure that they were involved in examining their own underlying beliefs about assessments and making changes in their classroom practice in alignment with their new understandings.

*3.4. Data Analysis*

A thematic analysis was conducted to analyze the three adopted sources of data collection: post-PD surveys, weekly reflections, and one-on-one interviews. At first, the post-PD survey and data collected from the PD reflections were read and analyzed. After conducting preliminary coding of input from the post-PD surveys and reflections, the interview transcripts were read multiple times, and the author started by assigning codes to each input using an inductive approach [43]. Assigned codes were based on the participants' responses rather than direct links to the specific question asked [43]. These steps are important to establish reliability of codes before identifying the emergent themes [42]. Data from the three sources of information was triangulated, and the author adopted the constant comparison method [36] to identify recurring data, create categories, and then systematically compare these categories and group them into themes and sub-themes. To increase the reliability of resultant codes and themes, the researcher sent various interview transcripts and the list of codes to her coder colleague in to obtain agreement on the final codes. All identified themes and sub-themes were organized in Table 1 below. These themes provided evidence that sustainable PDs support teachers in their roles and hence enhance the learning experience.

**Table 1.** Themes and sub-themes.

| # | Themes | Sub-Themes |
|---|--------|-----------|
| 1 | Changes in teachers' beliefs and attitudes about assessment practices | 1-Knowledge and skills gained<br>2-Practical application of knowledge<br>3-Increased confidence in designing assessments |
| 2 | PD impact on teachers' personal and professional growth | 1-Leadership skills<br>2-Collaboration skills<br>3-New teaching strategies |
| 3 | Sustainability of PD | 1-Application of knowledge<br>2-Reflective practice<br>3-Sharing knowledge and expertise<br>4-Empowerment of innovation<br>5-PD Impact on students' learning and critical thinking<br>6-Adopted PD modalities |
| 4 | Increased motivation and passion for teaching | ———————— |

## 4. Results

Data collected from the post-PD surveys, reflections, and one-on-one interviews were combined and analyzed based on a thematic analysis that generated a total of 4 main themes and 12 sub-themes, as detailed below.

*4.1. Theme 1: Changes in Teachers' Beliefs and Attitudes about Assessment Practices*

Data collected revealed the PD changed participants' perspectives about assessment practices. This was mainly due to *the knowledge and skills that they gained* in terms of integrating assessments with the lesson and designing rubrics to increase reliability of assessment results. As one female participant from Syria revealed:

> This PD taught me how to prepare a lesson plan by planning about the assessment first (backward design) and how to design rubrics as I never did this previously in my school.

Similarly, based on the data gathered through the post-PD reflections, participants mentioned various technical skills and knowledge they gained through the PD sessions. As one female teacher from Egypt mentioned in her reflection:

> I learned to use more than one assessment to collect data about students' learning and to improve their academic needs. I also learned the importance of collaborating with other teachers and asking for their opinions about the assessment that I am using to see how I can improve them by increasing their reliability, validity, and effectiveness.

Based on the results of the conducted interviews, participants highlighted gaining much knowledge and many skills related to evaluation and assessment, especially learning techniques and applications that can be used with students. As such, teachers' beliefs about assessment changed and improved as a result of the *practical application of knowledge* that this PD offered. As one Syrian participant shared:

> I benefited a lot from this PD, especially from the activities that helped us to apply the knowledge and how to create a rubric.

Consequently, the majority of the teachers participating in this PD reported that it *increased their confidence when designing assessments*. In particular, novice teachers benefited the most as they felt that they were not well-prepared or adequately trained in assessing students' learning when they first started their teaching journey. The Syrian female participant shared her thoughts through one of her reflections by saying that:

> This PD increased my confidence, especially since when we were hired as new teachers, we immediately started without any previous preparation or PD, especially that we were from different backgrounds and educational context.

Similarly, a more experienced teacher reported benefits similar to that of her novice colleague:

> I think now I became more confident in creating rubrics and designing assessments. I also learned the importance of self-assessment to improve students' reading and writing skills.

*4.2. Theme 2: PD Impact on Teachers' Personal and Professional Growth*

Based on the post-PD surveys, 64% of participants expressed their great satisfaction with the degree to which the PD improved their technical skills and helped them grow further professionally. Results of the conducted interviews corroborated the survey results as participants reported gaining lots of knowledge and skills relating to evaluation and assessment, especially learning new assessment practices they can use with students. As part of their personal growth, participants highlighted *the impact of this PD on their leadership skills as it* helped them to feel more at ease and in control within the classroom. As one participant from Egypt mentioned in her reflection:

This PD increased my leadership skills in my classroom, and I started depending more on my students and giving them more leadership opportunities and decisions to make regarding how and why they were being assessed.

Furthermore, participants highlighted that they were excited to train others with the information that they gained about assessments, as indicated by a male Syrian participant with over 10 years teaching experience: "Of course, just sharing the knowledge with others and standing up in front of other teachers increases my leadership skills." Other participants mentioned that the training increased his confidence and leadership skills as he initiated a new project at his department to revisit the formative assessments currently used. Additionally, the majority of participants (80%) explained the *impact that this PD had on their collaboration skills* with their colleagues and other peers. As clearly stated by one female participant who has less than 10 years of teaching experience:

Honestly, the activities that we did in the PD sessions were very beneficial as they were a great opportunity for us to apply the knowledge and create a healthy sense of competition, which encouraged us to collaborate in order to present the best work.

Another male participant from Syria explained that collaboration was the result of the strong relationships that were built among participants in the PD sessions:

This PD increased my relationship with other teachers at the same school and showed that we are one family, and that we collaborate together to support students' academic needs and deliver the knowledge in the most effective way.

Furthermore, it was highly evident that the impact of the PD on *teachers' teaching strategies* and pedagogies as they learned how to design assessments that engage students put them in real-life situations, and increased their higher-order thinking skills. As stated below by a male participant from Palestine with 9 years of teaching experience:

This PD taught me how to create a rubric, which helped me to have a structure for the assessment; and I felt that students quickly got used to the rubric and felt motivated and more engaged in the assessment.

It is noteworthy to mention here that the majority of participants (87%) reported in the high stakes assessments which are mandated from the Ministry of Education that, the overwhelming curriculum, the large class size, and lack of human resources were the primary restraints to implement real changes and hinder teachers' professional growth. The statements below support this finding:

I think that the changes I suggested to have in the assessments are highly effective but I know that they won't be implemented as I am accountable for covering the curriculum and ensuring that students did all high stakes assessments of the ministry.

(Week 9—reflection 3)

I am happy practicing this new assessment technique with my students and I can see how much students were engaged but I could not finish it and see the students' results as I have 30 students in my class and no teacher assistant.

(Week 9—reflection 8)

I could not implement the new way of assessing my students this week as I had to cover the learning outcomes as per the curriculum and I had no time left to be creative or use authentic formative assessments as the trainer taught us in the sessions.     (Week 9—reflection 11)

What we are learning here at this PD and other PDs is very effective, and I know that if we (teachers) implement the new strategies, we will improve the performance of the students but let's be realistic and look at the number of students inside each class.

(Male participant from Algeria, with less than 10 years of experience)

*4.3. Theme 3: Sustainability of PD*

The sustainability of any PD is related to the trainees' ability to transfer knowledge to their peers and *apply the knowledge* they received during their day-to-day work activities. As the PD sessions were given once a week, teachers had the chance to practice what they learned in designing assessments and rubrics, and analyzing assessment data. The majority of participants (87%), in fact, had the chance to do that and reflected on their practices in the weekly reflections. Furthermore, based on the results of the post-PD survey, 93% mentioned that they were very satisfied with the relevance of the PD workshops to their work. Data from the PD reflections also indicated that participants expressed various ways that they could apply what they learned in their work with students, in particular. The statements below support this finding:

"I will involve students with the type of assessment they are going to have."

(Week 1—reflection 12)

"I will use small assessment tasks that boost students' capabilities."

(Week 1, reflection 9)

"I will definitely include critical thinking questions in my assessments. Also, I will share important issues with students and give them time to find out more about the topic; we will work together to give suggestions and solutions."

(Week 5, reflection 2)

Another characteristic that made this PD sustainable is *the reflective practice* that participants were involved in during the PD time. Based on the results of the interviews, participants named various ways they can use the knowledge and skills they learned during the PD and highlighted the important role that the reflections played in making them think and come up with ways to use this knowledge in their daily work. Furthermore, data collected showed that the relationship built between participants in the sessions increased their capabilities to *share knowledge and expertise* among them, and with their colleagues at school. During the interviews, participants mentioned that they would share the content of everything they learned with their peers, explain the methods to be used, and further highlighted that the trainers' delivery of the PD would help them in doing so. As one participant indicated:

I will conduct 10 workshops same as what the trainer did with us, and I will ask the same reflections as the trainer asked us to fill and to participate in all the activities that we did. I will do this so that potential participants will go through the same experience that I went through.

(Female participant from Egypt, with 8 years of experience)

I used to take the knowledge that I gained from each session to my team in the math department, and they applied it. For example, when students plan the content of the subject, this helped them to remember the concepts.

(Male participant from Palestine, with 9 years of experience)

It is noteworthy to also mention that the PD *empowered participants to suggest innovative ideas*, another characteristic of sustainability in the context of this study. According to interviewed participants, the PD encouraged them to share innovative ideas about assessments and propose them to the school leadership team. The statements below support this finding:

I suggested to the coordinator of the Arabic department that the current assessment questions should be modified by including more open-ended questions that would entail students to demonstrate more critical thinking in their answers.

(Male participant from Jordan, with more than 10 years of teaching experience)

> This PD encouraged me to suggest many new creative ideas in assessing and evaluating students' learning to help them, especially the weaker ones. I want to use these creative ideas to increase the confidence of weak students as they feel embarrassed when they hear their scores on the exams in front of other students.
>
> (Female participant from Syria, with more than 10 years of experience)

Another characteristic that emerged from data collected that identified the sustainability of this PD is related to the overall outcome, which is to *improve students' learning and critical thinking*. While this sub-theme could be hard to determine as this study did not collect any data related to students' performance before or after the PD was conducted, participants' reflective notes and their responses to the surveys and interviews showed that the PD had a positive impact on their teaching pedagogies. As one participant from Algeria stated in his reflection, "This PD guided me on how to become a facilitator to students' knowledge and guiding them, and to create a more student-centered classroom" (week 3, reflection 4). Another female participant from Egypt mentioned, "I learned the importance of self-assessment to improve student learning, and I started using this technique since the initial sessions of this PD, especially in teaching reading and writing skills." After the participating teachers started to apply the skills and knowledge they learned from the PD within their classrooms and modified how they dealt with the students, they mentioned that the student's critical thinking skills, confidence, debating skills, and ability to assess the performance of other students improved a lot. The statements below support this finding:

> From the first time I used a rubric with my students, their critical thinking started to increase. I also learned [about] creating authentic assessments that would think, analyze, and evaluate knowledge and information.
>
> (Female participant from Egypt, with 8 years of experience)

> I used another assessment strategy that the trainer taught us in the PD sessions, which is the pros and cons; this strategy gave students a voice and helped them express and justify their opinions. This has also increased their debating skills.
>
> (Female participant from Egypt, with eight 8 years of experience)

> I am very happy that the students developed their own opinion now and when I discuss a topic with them, they are able to argue using logic."
>
> (Female Egyptian, with more than 10 years of experience)

What also made this PD sustainable is its *adopted modalities* in both face-to-face and online settings. Participants showed a high level of attendance in both modalities; however, the vast majority (93%) reported in the interviews that they preferred the face-to-face modality. They mentioned that this modality helped them to build stronger relationships with other attendees, better communication, and more engaging discussions. The statement below supports this finding:

> I liked face-to-face PD more because it showed our facial expression, the environment, and the engagement. The environment impacted a lot the engagement of some teachers as they didn't have access to a quiet place at their school without noise.
>
> (Female participant from Jordan, with more than 10 years of experience)

The many different facets of sustainability in the findings point to the need for PDs to support it and ensure that improved learning and higher quality education take place through changes in the way teachers, students, and managers think and learn.

*4.4. Theme 4: Increased Motivation and Passion for Teaching*

On another note, the PD increased *participants' motivation and revived their passion towards teaching.* This is important as it is associated with the degree of effectiveness of PDs generally and in the teaching profession specifically.

This PD increased my passion for teaching, introduced me to new experiences, and corrected the wrong information I have about assessments.

(Female participant from Syria, more than 10 years of experience)

Increasing teachers' motivation and passion for teaching is a significant theme to be considered here as the majority of teacher participants are experienced with some working at the same school for a long time. That being said, this PD not only reached its aims in developing teachers' assessment practices, but it also revived their passion for teaching and increased their motivation to engage students in assessments and to try new teaching strategies that would encourage them to experiment and eventually make some changes in their classroom practices.

## 5. Discussion

Findings of this study showed that teacher participants' beliefs and attitudes about assessments changed dramatically for novice and more experienced teachers equally. Sustainable PDs, therefore, offer teachers a sure way to make not only their roles more rewarding, but through such initiatives students of all abilities can be catered to in learning and assessment. Participants' confidence in creating assessments and rubrics and in engaging students in self-assessment increased. This finding clearly shows that participants of this study have the capacity to make positive changes in their classroom practices, which would positively impact their students' learning outcomes [11,12]. The UN's sustainability goal for 2030 [1] of endorsing quality education that promotes equality and equity for lifelong learning can be achieved through sustainable specific PDs such as the one described in this paper. For education to offer more quality experiences to children, teacher training needs revision through mainly placing the teacher at the center of development and offering them meaningful agentic roles.

It should be noted, though, that although the current PD was successful, the large class size, lack of teaching assistants, overwhelming curriculum, and structure of the low and high stakes assessments at the public education sector in the UAE restricted teachers' agency to affect any meaningful changes. This reflects previous educational research recommending alignment between the ministry's vision, policies, and goals to be achieved while keeping in mind the resources, curriculum, and teacher [3,44]. If teachers feel that the knowledge reaped from PDs would not be implemented and used to make a change in the classroom, which is the case here, they would lose confidence in their role and their abilities to be creative and innovative in their teaching practices. Similar to the recommendations made by [3] for policymakers in Qatar, the teaching hours and autonomy of teachers in the UAE must be revisited if sustainability and its benefits are to be felt by both students and teachers. Nevertheless, participants of this study appreciated the application of authentic and engaging assessments to meet diverse students' needs and develop their higher-order thinking skills. By applying the knowledge gained from the PD module in their classroom, experimenting with it, and reflecting on its use and effectiveness, participants were able to provide their students with engaging sustainable assessments that can develop higher-order thinking skills. Another weakness of this study is that, despite this finding, we cannot assume that this PD had a direct impact on students' outcomes as (1) it did not measure students' outcomes, and (2) teachers need regular PDs to enhance their ability to improve their testing tools and impact students' outcomes [45].

While incidental learning was shown to benefit teachers' professional learning, the majority of PDs, including this one, are thoroughly designed based on teachers' needs. This is fundamental as they engage participants in experiential learning through developing their practices in action, experimenting with these practices, and reflecting on them [16]. The PD provided to participants of this study motivated them to implement the knowledge they gained in their classroom and encourage their colleagues to experiment with these new practices. However, we argue similarly to [25] that the social interaction that this PD provided to teachers created a community of practice as they built relationships and connected with one another outside the PD hours. Both male and female teachers had a

similar positive attitude towards this PD, as not only was it identified as per their needs [14] but it also rekindled their passion for teaching.

Findings of this study showed that the design and implementation of this PD made it sustainable in many ways. First, it engaged participants in experiential learning, which is believed to impact individuals' professional and personal growth [16], particularly their leadership and collaborative skills. Being able to transfer knowledge to others and design and conduct PDs for their colleagues provided teachers with a sense of gratification, which would eventually help them become agents of change [25]. This has also empowered them to take initiatives and suggest innovative ideas about assessment design and implementation. To this end, participants felt that their confidence was enhanced and leadership skills inside and outside the classroom were developed. This result is significant to evaluate the effectiveness of this piloted sustainable PD [13,25].

Second, this PD is sustainable as it gave participants activities to practice the knowledge in real experiences, and enough time to experiment and reflect on their practice as they were asked to implement what they learned in their classroom, and reflect on its success or failure [7]. PDs that are inspired by experiential learning are believed to have a direct impact on making a change in individual beliefs [18]. In the context of this study, participants reported that their assessment practices changed as they were able to create activities in class that involve students in designing rubrics for the assessments. Furthermore, this PD is sustainable as it proved to be adapting to changes and resistant to disrupting factors [37]. This was seen, in particular, when mid-way through the PD, sessions had to be switched from face-to-face to online. Although participants of this study preferred the face-to-face mode, it did not impact their attitudes or make a difference on the benefits they reaped regarding their professional growth and leadership skills. This finding is in alignment with previous findings of [46] that found no difference on teachers' knowledge and beliefs, their classroom practice, or student learning outcomes related to PD modality. The paper calls for more sustainable teacher PD programs as a positive response to the UN's 2030 sustainable development goal of quality education.

## 6. Limitations and Recommendations

This piloted sustainable PD tackled several factors such as leadership, the chance to practice and exchange expertise, and recognition of teachers' performance through the certificates that were distributed at the end but also through the leadership roles they took as trainers in their own school setting, and allocation of time to give the opportunity for those teachers to practice what they learned and expand their professional orientation [7]. However, we cannot assume that it has had a direct impact on student learning. Further studies interested in evaluating the effectiveness of PDs should collect pre- and post-data about students' performance, as this is considered a significant variable [10]. Furthermore, it would also be beneficial to evaluate the effectiveness of this PD module by monitoring the efficacy of sharing best practices and the exchange of expertise that was carried out by participants with their colleagues. Since all participants were teaching at the same type of schools that follow the same curriculum and same teaching practices and assessments, it would be significant to evaluate the impact of this PD module on teachers' attitudes, professional growth, and leadership practices, since they are teaching at different types of schools and curricula. Although many PDs are conducted yearly in schools around the UAE, it would be better if these were based on teachers' interests and needs to yield the desired results when it comes to improving student learning and achievement. It is equally important to revisit the structure and policies of students' assessment practices in the public sector to align with what the reform is claiming to change and improve. Lastly, findings of this study could be used as scientific data to create a model for developing sustainable PD educational programs in the UAE. To that end, we recommend that practitioners design PDs that are based on the criteria identified in this study to ensure the sustainability of these programs.

## 7. Conclusions

Professional development can be a powerful tool for improving teachers' efficacy and thus the students' learning. However, there is a need to develop new approaches to teacher learning that create real changes in teacher practice and improve student achievement. With the increasing demands from the education sector in the UAE to meet the expectations of the UAE vision 2030, this study explored the impact of a sustainable PD on developing teachers' attitudes about assessment practices, professional growth, and leadership skills. Since the PD examined in this study was designed to meet the needs of teachers in the UAE, the participants in their different positions not only benefited from the information they gained about conducting proper assessments to improve student learning but also demonstrated professional and personal growth. Although many PDs are conducted yearly in schools around the UAE, it would be better if these were based on teachers' interests and needs to yield the desired results when it comes to improving student learning and achievement. It is equally important to revisit the structure and policies of students' assessment practices in the public sector to be aligned with what the reform is claiming to change and improve.

**Funding:** This research was funded by Zayed University, RIF grant number R21112.

**Institutional Review Board Statement:** The study was conducted in accordance with the Declaration of Helsinki, and approved by the Institutional Ethics Committee of Zayed University protocol code ZU22_902_Yon April 2022.

**Informed Consent Statement:** Informed consent was obtained from all subjects involved in the study.

**Data Availability Statement:** Data collected including interview recordings and transcripts, surveys and reflections can be shared by the author to MDPI as per their request.

**Acknowledgments:** The author would like to acknowledge her colleague Fatma Said for proofreading the paper.

**Conflicts of Interest:** The author declares no conflict of interest. The funders had no role in the design of the study; collection, analyses, or interpretation of data; writing of the manuscript; or decision to publish the results.

## Appendix A. Interview Questions

Dear Participant,

Thank you for agreeing to take part in this interview. The aim of this project is to collect data about your experiences in the PD and learn more about the impact of this PD on your beliefs and attitudes towards assessments, and also towards your professional growth. This interview will last for approximately 30 min during which you will be asked a total of 18 questions. You must be assured that your participation will remain both anonymous and confidential. You will receive a copy of the interview transcription for revision to ensure transparency and trustworthiness. The research results will be shared with you as well.

I would like to take the opportunity to thank you for your cooperation.

Best Regards,

Dr. Sandra Baroudi

1. Why did you join this module?/expectations?
2. What was the best thing you liked in this module? Why?
3. What are the challenges or barriers you faced in this module?
4. How did you overcome these challenges?
5. What was your overall experience in this module?
6. What changes or recommendations would you suggest?

   Knowledge and skills

7. To what extent has this module increased your knowledge and skills in assessment? How?
8. How would you apply the information gained from the module to generate new ideas or innovate when assessing students' learning?

9. How would you apply information gained from the module to increase students' critical thinking through assessment?

   Impact on students' outcomes

10. How did this module change the way you think about assessment?
11. Do you agree with the statement "teachers use assessment to increase students' learning and not to label them?"
12. How are you going to apply what you have learned in your classroom (future plans)?
13. Based on the training you received, can you explain how you will use the assessment to improve your students' outcomes?

    Personal growth and Leadership practices

14. Do you feel that this module increased your confidence in assessing students' learning? How?
15. How did this module impact your leadership practices in your classroom or school-wide? Motivation? Feelings? Build relationships with other colleagues?
16. Did this module help you build relationships with others? How?
17. Do you feel you are now more prepared to take on new roles related to assessment (i.e., designing assessments/rubrics, collaborating with teachers, etc.)?
18. Now do you feel you are willing to take risks in the way you assess students? How?

## Appendix B. Survey Questions

Dear participant, Thank you for agreeing to fill in this survey. The aim of this project is to collect data about your experiences during the PD sessions and learn more about the impact of this PD on your beliefs and attitudes towards assessments, and also towards your professional growth. The survey will take 10 to 15 min of your time. The first part of the survey included eight questions to collect participants demographics; the second part included 19 questions based on 5-Likert scale ranging from very satisfied to very dissatisfied; and the third part constituted three open-ended questions to understand the challenges, if any, that the participants faced regarding the location of the PD, the timing, the topics of the sessions, and their recommendations. You must be assured that your participation will remain both anonymous and confidential. Please note that the research results will be shared with you as well. I would like to take the opportunity to thank you for your cooperation.

Best Regards,
Dr. Sandra Baroudi
Part A—Demographics

1. Gender    Male    Female
2. What is your nationality?
3. Age range    25–29    30–39    40–49    50–59    60+
4. Years of Experience    less than 5    5–10    10–15    more than 15
5. Years' experience at the charity schools    less than 5    5–10    10–15    more than 15
6. Years in your current role    less than 5    5–10    10–15    more than 15
7. Educational qualification    Bachelor    Master    Doctorate    Other, specify
8. What subject do you teach?

   Part B—Likert scale

   Please rate the following questions from: Very satisfied—Satisfied- Neutral- Dissatisfied Very dissatisfied

1. The PD module sessions were well organized.
2. The objectives of the sessions were clear to me.
3. The topics covered during the sessions were relevant to my daily practice.
4. The facilitator provided high-quality training.
5. The deliverables of the sessions were clear.
6. I received ongoing support during the session.

7. The resources and materials used in the sessions were of a great support.
8. The location of the sessions was convenient and accessible.
9. The timings of the sessions were convenient and accessible.
10. The sessions met my needs as an educator.
11. The sessions impacted my professional practice
12. The sessions offered me practical skills and information for me to apply in the classroom.
13. The sessions were designed to challenge participants with different years of experience.
14. The sessions were designed to challenge participants teaching at different school levels.
15. The sessions were designed to challenge participants teaching different subjects.
16. The sessions helped me prepare for my career advancement.
17. This PD module strengthened my confidence in designing assessments for my students.
18. This PD module prepared me to take more leadership posts at my school.
19. I would recommend this PD module to my colleagues.

Part C—Open-ended questions

1. What did you like the most about this PD module?
2. What improvements would you recommend?
3. Is there anything else you would like to share?

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
