# Peer review of "Exploring Teacher Education for Sustainable Development in the UAE"

_sustainability, doi:10.3390/su15031981_

Round 1

Reviewer 1 Report

The manuscript entitled “Exploring Teacher Education for Sustainable Development in the UAE” is a valuable and engaging contribution to the literature on a type of pedagogy that can make a difference. The author’s goal is to assess the subjective efficacy of a novel professional development program for the education of teachers. The evidence collected is qualitative. It supports the predicted effectiveness of the novel program based on 4 main areas/themes.

In the introductory section or literature review, a more in-depth review of the concept of sustainable professional development programs is advised. In my modest opinion, the definition of such programs needs to be augmented with some practical examples. Most importantly, practical examples are likely to shed light on the fact that conceptualizations of sustainable professional development programs may vary, especially in their implementations.

 A specific hypothesis for each theme and its rationale may need to be described. For instance, why would this novel program be expected to change participants’ motivation and passion for teaching? How and why would the program be expected to change teachers’ beliefs and attitudes about assessment practices?

In the method section, briefly define the differences in responsibilities between a teacher and a head teacher. In addition, explain the rationale for selecting both teachers and non-teachers.  

A more complete description of the program needs to be put forth. What are the specific program learning outcomes? How are learning outcomes measured? What are the specific assessment measures that are used to ensure that the program is indeed capable of achieving particular targets (i.e., changes in teachers’ beliefs and attitudes about assessment practices, ability to impact teachers’ personal and professional growth, sustainability of PD, and increased motivation and passion for teaching)?

The discussion section may be amplified by covering more broadly the purported differences between existing professional development programs and the current one. Interestingly, one of the limitations of the study is its lack of objective assessment of teachers’ performance. Is this limitation a weakness or a negligible property of the study?  Are there other limitations that may need to be explored? Consider the issue of generalizability of the current findings. Can the sample of participants be considered prototypical of teachers in the UAE?  To what extent can the findings apply to teachers in other Gulf States or the broader Middle East? Are there cultural and structural differences that may require consideration? 

Author Response

Dear reviewer, I would like to thank you for your valuable comments. Please see each point addressed in the file attached and also highlighted in yellow in the revised mansucript. 

Thank you so much 

Reviewer 2 Report

1. In the study, the researcher states "This paper calls for the development of sustainable PDs. But why would you choose teachers? Please use citations to support the point.

2. The literature review is too general and should be written under subheadings according to the research hypothesis.

3. In the materials and methods, please add section of research setting.

4.Please add a section to introduce the questionnaire and interview.

5. Please introduce the reliability and validity of the thematic analysis.

6. I recommended that the qualitative and quantitative results be presented together.

7. The discussion lack the section of research limitations and suggestions for future research.

8. Please add a chapter of conclusion. The results of the study lack the theoretical knowledge. I suggest to add it.

9. Please add more references for 2023, 2022 and 2021.

Author Response

(The authors gave the same response as above.)

Round 2

Reviewer 2 Report

Please do not consider the word limit. I suggest adding a conclusion.

Author Response

Dear Reviewer, 

Thank you for your valuable comment below. A conclusion section is added as per your request. 

Thank you